# Biological Modulation of Autophagy by Nanoplastics: A Current Overview

**DOI:** 10.3390/ijms26157035

**Published:** 2025-07-22

**Authors:** Francesco Fanghella, Mirko Pesce, Sara Franceschelli, Valeria Panella, Osama Elsallabi, Tiziano Lupi, Benedetta Rizza, Maria Giulia Di Battista, Annalisa Bruno, Patrizia Ballerini, Antonia Patruno, Lorenza Speranza

**Affiliations:** 1Department of Innovative Technologies in Medicine and Dentistry, University G. d’Annunzio, 66100 Chieti, Italy; francesco.fanghella@phd.unich.it (F.F.); dibattistamariagiulia@gmail.com (M.G.D.B.); annalisa.bruno@unich.it (A.B.); patrizia.ballerini@unich.it (P.B.); 2Department of Medicine and Aging Sciences, University G. d’Annunzio, 66100 Chieti, Italy; sara.franceschelli@unich.it (S.F.); valeria.panella@unich.it (V.P.); osama.elsallabi@unich.it (O.E.); tizianol.lupi@phd.unich.it (T.L.); benedetta.rizza@phd.unich.it (B.R.); antonia.patruno@unich.it (A.P.); lorenza.speranza@unich.it (L.S.)

**Keywords:** nanoplastics, autophagy, mitophagy, mTOR, cardiovascular system, gastrointestinal system, nervous system, reproductive system, respiratory system

## Abstract

Nanoplastics (NPs), an emerging class of environmental pollutants, are increasingly recognized for their potential to interfere with critical cellular processes. Autophagy, a conserved degradative pathway essential for maintaining cellular homeostasis and adaptation to stress, has recently become a focal point of nanotoxicology research. This review synthesizes current evidence on the interactions between NPs and autophagic pathways across diverse biological systems. Findings indicate that NPs can trigger autophagy as an early cellular response; however, prolonged exposure may lead to autophagic dysfunction, contributing to impaired cell viability and disrupted signaling. Particular attention is given to the physiochemical properties of NPs such as size, surface charge, and polymer type, which influence cellular uptake and intracellular trafficking. We also highlight key mechanistic pathways, including oxidative stress and mTOR modulation. Notably, most available studies focus almost exclusively on polystyrene (PS)-based NPs, with limited data on other types of polymers, and several reports lack comprehensive assessment of autophagic flux or downstream effects. In conclusion, a better understanding of NP–autophagy crosstalk—particularly beyond PS—is crucial to evaluate the real toxic potential of NPs and guide future research in human health and nanotechnology.

## 1. Introduction

The invention of plastics has been one of the most significant innovations of the last century, as plastics have had a profound impact on numerous industries, especially in producing of everyday items and food packaging [1]. These materials are synthetic and made from long chains of organic molecules with different chemical groups [2]. A chemical process known as “polymerization” creates plastics, where small molecules, known as “monomers”, unite to form long chains. Plastics can be produced in several ways, depending on the monomers used [3]. Commercially used polymers are always formulated with additives such as stabilizers, flame retardants, plasticizers, fillers, and pigments, which contribute to the overall materials’ properties [4].

According to Abdiev et al. (2022), plastics are highly popular in daily usage due to their lightness, flexibility, strength, and chemical and corrosion resistance. The demand side of modern life is also covered by their low cost and ease of production [5]. The wide range of artificial polymers has been referred to as a “family of four”, specifically: (1) thermoplastic polymers or “plastics”; these are the variety of the polymers that soften when they are heated beyond a specific temperature and reshape before solidification upon cooling; (2) thermosetting polymers or “thermosets”, provided in forms of precursors (mainly liquid) and cured via a chemical reaction after shaping into an infusible, insoluble polymer network; (3) elastomers; and (4) synthetic fibers [6].

The world’s plastic production began in the 1950s [7] and has continuously increased due to its widespread use in everyday life. Although the world’s plastic production was 1.5 million tons in 1960, it climbed as high as 335 million tons in 2016 [8]. In 2010, only 19 coastal countries produced 275 million tons of plastic [9]. The leader in terms of demand is the packaging sector, which comprises 40.5% of the total production, followed by civil construction (20.4%) and the automotive industry (8.8%). This situation results in ever-increasing plastic waste generation. As reported by Geyer et al. (2017), from 1950 to 2015, plastic waste increased to 6.300 million tons. Of this waste, the United States is the largest contributor, with 37.729 tons per day, followed by China (31.665 tons per day), Japan (19.606 tons per day), and Brazil (12.272 tons per day) [10].

Due to improper waste disposal, management deficiencies, and the slow degradation of polymers, plastic waste can persist in the environment for extended periods, potentially requiring 20 to 450 years to degrade [11,12]. Therefore, it is not surprising that contamination occurs in terrestrial environments, particularly in areas with high anthropogenic activity such as urban and agricultural zones. However, in recent years, scientists and the public have become increasingly concerned about the adverse effects of plastics on the environment and human health [1]. When plastics enter the environment, they slowly break down into smaller pieces known as “microplastics” (MPs), which are smaller than 5 mm, and “nanoplastics” (NPs), which are smaller than 1000 nanometers [13].

Since then, NPs have been the subject of extensive research, which enabled insight into their pleiotropic properties. The role of NPs on autophagy has been studied only recently, particularly during the last 3 years, in which about 80% of the manuscripts have been published. However, these recent studies have not yet been reviewed. Given the emerging importance of NPs and autophagy in mediating the detrimental effects of this new toxic pollutant, this review provides an update on studies focused on the relationship between NPs, biological systems, and the autophagy process.

It is well recognized that both MPs and NPs can interfere with cellular pathways. However, due to their specific physicochemical properties, including nanoscale size, high surface reactivity, and enhanced cellular internalization, NPs can interact more profoundly with intracellular processes. Thus, this review aims to provide an up-to-date overview of the effects of NPs on the modulation of autophagy. Importantly, since the research in this area is still emerging compared to the extensive literature on MPs, a focused analysis of NPs is essential to highlight current knowledge gaps and address future investigations in the field.

## 2. MPs and NPs: Properties, Environmental Fate, and Biological Impact

### 2.1. Classification, Characteristics, and Forms of Microplastics

MPs are solid plastic particles that are insoluble in water. They range in size from 1 µm to 5 mm and can have different shapes [14]. The amount of MPs pollution is connected to the use of common plastics like polyethylene (PE) of high density (HDPE) and low density (LDPE), polypropylene (PP), polystyrene (PS), polyvinyl chloride (PVC), polyethylene terephthalate (PET), and polymethylmethacrylate (PMMA) [15,16]. Polymers are still mainly derived from petroleum-based raw materials [17] (Table 1).

MPs are very stable and do not break down easily, so they can stay in the environment for a long time—in water, soil, and air. Because they have a large surface area, they can attract and carry other harmful chemicals and microorganisms [18]. Their ability to transport toxic substances, which may cause cancer, genetic damage, or hormone problems in animals and humans, makes them a danger [19].

MPs are classified as primary and secondary [20]. Primary MPs are made small on purpose and are used in products such as cosmetics, soaps, scrubs, and medical devices. For example, these small beads or microspheres are often used as alternatives to natural exfoliants [21]. Secondary MPs originate from degradation of bigger plastic items such as bags, bottles, or fishing nets [22].

There are five common types of MPs: fragments, spheres, fibers, industrial granules or pellets, and foams [23]. MPs can look very different. For example, microfibers are released from clothes during washing. Fragments come from broken plastic objects [24]. Microfibers last longer than other types, like films or foams. Their appearance can change after exposure to sunlight, heat, or moisture [25].

Ultraviolet rays (UV), physical damage, and biological activity break these down over time [26,27]. These MPs typically have irregular shapes and rough surfaces [28] and are released through weathering, microbes, and movement from soil or sediments [29]. Below is a description of the processes leading to the formation of secondary MPs and NPs.

**Table 1 ijms-26-07035-t001:** Classification, characteristics, and forms of microplastics.

Category	Description	Common Polymers	Origin and Formation Mechanisms	Morphology and Physical Features	Environmental and Health Implications	References
**Definition**	Solid plastic particles, insoluble in water, ranging from 1 µm to 5 mm in size	PE, HDPE, LDPE, PP, PS, PVC, PET, PMMA	Primarily derived from petroleum-based feedstocks	Diverse shapes including spheres, fragments, fibers	Persistent pollutants with potential for long-term environmental accumulation	[14,15,16,17]
**Primary Microplastics**	Manufactured intentionally at microscale for specific applications	Microbeads in cosmetics, scrubs, medical devices	Industrial production of microscale plastic particles	Typically spherical, uniform size distribution	Direct environmental release; potential for bioaccumulation in aquatic and terrestrial organisms	[20,21]
**Secondary Microplastics**	Result from the fragmentation and weathering of larger plastic debris	Derived from consumer products such as plastic bags, bottles, fishing nets	Fragmentation driven by photodegradation (UV radiation), mechanical abrasion, and microbial activity	Irregular shapes, rough and heterogeneous surfaces	Act as vectors for toxic chemicals and pathogens; increased bioavailability and toxicity potential	[22,26,27,28,29]
**Stability and Persistence**	Highly resistant to biodegradation and environmental breakdown	All aforementioned polymers	Chemical inertness and polymer structure confer environmental persistence	-	Long residence times in soil, water, and atmospheric compartments; challenges for waste management	[18]
**Surface Properties**	High specific surface area facilitates adsorption of pollutants	-	Surface properties promote sorption of hydrophobic organic contaminants and microbial colonization	-	Enhanced transport of hazardous substances; potential to disrupt biogeochemical cycles	[19]

### 2.2. Human Exposure Routes

People can be exposed to MPs and NPs through inhalation, ingestion, and skin contact [30]. The most common way for MPs and NPs to enter the body is by ingesting contaminated products. These particles have been found in everyday foods, like salt, sugar, honey, milk, fruits, soft drinks, and even beer and water [31]. It is estimated that each person ingests between 39,000 and 52,000 plastic particles per year through food [32]. The gut is the first organ that faces these particles when swallowed and absorbed. Several studies have shown how MPs and NPs can damage the gut barrier, alter the microbial balance, and cause mild inflammation and oxidative stress in healthy people [33,34,35,36]. Still, these effects are usually insufficient to trigger a real intestinal disease [37,38]. This suggests that a healthy gut may resist the presence of these particles [39]. However, more and more research points to the possibility that MPs and NPs might worsen some diseases, like obesity, liver damage caused by alcohol, or gut inflammation [40,41,42]. Additionally, damage to the intestinal barrier is often associated with more serious conditions.

Inhalation is another method by which MPs and NPs can enter the human body. Some studies suggest that we might inhale 3 to 15 times more MPs than we ingest [31,43]. These particles come from many sources. Synthetic clothes can release fibers. When plastic materials wear down, they can also produce MPs. Other sources include building materials, plastic waste, landfills, and incinerators. These particles are found in outdoor and indoor air [44]. Inhalation is now recognized as one of the primary ways plastic particles enter our bodies. Whether or not a particle is inhaled depends more on its aerodynamic equivalent diameter (AED) than on its actual size. This is because AED considers a particle’s shape, density, and surface, not just its size [45,46].

Contact with the skin is considered a less important way of exposure. However, skin cells can still be affected, mostly due to oxidative stress [47,48]. Some research has found both MPs and NPs on the skin, especially the face. If exposure occurs frequently, the small size of the particles may allow them to pass through the skin and cause different effects on the body [49,50].

Although dermal contact is a minor route, tiny plastic particles, like NPs under 100 nm, can still penetrate the skin. These particles may have toxic effects and should not be ignored. The outer layer of the skin, known as the “cuticle”, is the first protective barrier. It slows down or blocks the entry of most particles and chemicals, but this barrier is imperfect. Two main factors enable plastic particles to penetrate the skin: their size and surface properties. Smaller particles (less than 100 nm) with higher surface activity are more likely to pass through [51]. Larger particles may still enter the body, but usually do so through hair follicles, sweat glands, or minor cuts or injuries in the skin [52].

### 2.3. NP-Specific Toxicity and Cellular Mechanisms: The Importance of Protein Corona and Size

NPs are smaller than MPs. They can easily enter cells because they are so small and have a large surface area. Once inside, they might damage the cells, affect their activities, and even cause harm [53]. NPs can also act as carriers for harmful substances, like heavy metals and toxic chemicals. This makes them even more dangerous when they enter the body [8].

Once inside the body, both MPs and NPs can interact with cells differently. These interactions depend on many factors, including surface structure, particle size, and the extent to which they adhere to molecules in the body [54]. In biological fluids, NPs tend to bind with proteins. This forms a layer around them, called the “protein corona” [55,56]. This coating alters how the particles behave and how cells respond to them. It also makes them harder to track or remove from the body. The reason is that NPs can trap large molecules on their surface. This changes their chemical and physical nature, affecting how they move or stay in specific environments. These changes can also make the particles more harmful [57,58].

The corona’s composition depends on the particle’s shape, surface chemistry, and material [59,60]. This protein layer is involved in several biological processes. It helps decide where particles go, how drugs are released, how they enter cells, and how effective treatments might be [61].

There are two kinds of interaction between the protein corona and NPs: hard corona and soft corona [62] (Figure 1). The hard corona consists of proteins that bind strongly to the NP surface. This binding is stable and long-lasting, meaning that the proteins remain attached and are not easily replaced. In some cases, the replacement of these proteins takes more time than the nanoparticle needs to enter a cell. [63,64]. The soft corona is different. It is less stable and includes proteins that attach weaker. These proteins can bind directly to the NPs or on top of the hard corona. They can come off easier or be replaced when the environment changes. Usually, the protein corona is around 20–30 nm thick. However, this can change, depending on the NPs and the biological fluid around them [62]. The formation of hard or soft corona depends significantly on the affinity of proteins. A protein with strong affinity will bind faster and become part of the hard corona [65].

Another factor to consider is size, which is particularly important when discussing biological effects (Figure 1). For instance, Li et al. (2024) studied retinal cells, using ARPE-19, exposed to 2 μm MPs and found no significant changes in cell health or stress levels. However, when they used 50 nm NPs, the cells became less viable and showed signs of oxidative stress. These smaller particles also crossed the cell membrane, damaged mitochondria and activated a response called “mitophagy” [66].

The cellular uptake of particles occurs mainly through two mechanisms: passive diffusion and active transport [67]. Passive diffusion is typically driven by concentration or potential gradients and generally involves small particles [68]. These particles can directly interact with intracellular structures, potentially causing significant cytotoxic effects [69]. In contrast, active transport requires energy and involves specific pathways such as clathrin- and caveolae-mediated endocytosis. This mechanism is commonly used for the internalization of larger particles, which are engulfed by membrane invaginations, forming vesicles that are transported into the cytoplasm [70,71].

Recent studies have shown which cellular uptake of polystyrene nanoplastics (PS-NPs) is highly dependent on particle size. In HeLa cells, PS-NPs with a radius of 10 to 15 nm were efficiently internalized and observed within the cytoplasm, whereas 25 nm particles exhibited limited uptake. Larger particles—40 nm, 50 nm, and especially 500 nm—were mostly retained on the cell surface and did not show significant internalization. Notably, only internalized particles were associated with toxic responses, including decreased cell viability, increased production of reactive oxygen species (ROS), and altered gene expression linked to oxidative stress and apoptosis. These findings suggest that cellular internalization is a critical factor in determining the toxicity of NPs [72].

Further insights on the size-dependent uptake of polystyrene micro- and nanoplastics were provided by Liu et al. (2021). Using both artificial membranes and RBL-2H3 cells, the authors demonstrated that smaller particles (50 nm and 500 nm) were capable of adhering to the membrane via hydrophobic and van der Waals forces and were subsequently internalized. The 50 nm particles primarily entered cells through clathrin-mediated endocytosis, while the 500 nm particles used both caveolin-mediated endocytosis and macropinocytosis, with the latter being predominant. In contrast, larger particles (5 μm) were neither able to adhere to the membrane nor to be internalized, further confirming the key role of particle size in cellular uptake mechanisms [73].

It has been reported that toxicity increases as the size decreases due to a high surface-to-volume ratio, allowing organic and inorganic contaminants, viruses, bacteria, and other pollutants to be conveyed in different body districts [74].

This suggests that small sizes play a crucial role in toxicity, and for this reason, in vivo studies showed that PS-NPs (0.5–50 mg/kg bw for 7 days) significantly induced the increase in permeability of the blood–brain barrier (BBB), and dose-dependently accumulated in the brain of mice. Additionally, PS-NPs were found to be present in microglia, and to induce microglial activation and neuronal damage in the mouse brain [75].

### 2.4. Autophagy Process

Autophagy is widely acknowledged as a fundamental and highly regulated cellular process for maintaining cellular homeostasis and survival. It plays a pivotal role in several biological domains, including evolutionary development, immune response modulation, and the regulation of programmed cell death. Its dysfunction has been implicated in the onset and progression of various life-threatening diseases, such as neurodegenerative disorders, autoimmune pathologies, and a broad spectrum of malignancies. Autophagy exerts a dualistic function within the cell. On the one hand, it supports longevity and promotes cellular survival by eliminating damaged organelles and misfolded proteins, thus preserving intracellular integrity. On the other hand, under certain pathological or prolonged stress conditions, it can shift toward a pro-death mechanism, leading to autophagy-associated cell death [76].

Mechanistically, autophagy is a lysosome-dependent catabolic pathway primarily responsible for the degradation and recycling of dysfunctional organelles and intracellular waste. This process allows for the regeneration of essential biomolecules and the production of metabolic energy, facilitating cellular recovery and adaptation to stress. Impaired autophagy is typically marked by accumulated damaged organelles and cellular debris, which can contribute to cellular dysfunction and disease pathogenesis. Recent research has reinforced the concept that autophagy primarily serves as a survival mechanism under a range of cellular stress conditions, including endoplasmic reticulum (ER) stress, microbial infection, oxidative damage, and hypoxia [77].

There are three well-characterized types of autophagy: macroautophagy, microautophagy, and chaperone-mediated autophagy (CMA). Each of these pathways ultimately leads to the lysosomal degradation of cytosolic substrates, albeit through different cargo recognition and transport mechanisms. Among them, macroautophagy—often referred to as “autophagy”—is the most extensively studied and understood. It involves the formation of double-membraned vesicles known as “autophagosomes” that engulf targeted cellular components and subsequently fuse with lysosomes for degradation [77].

The regulation of autophagy is governed by a complex interplay of molecular signals operating at multiple levels. At the transcriptional level, several key transcription factors have been implicated in modulating autophagy-related genes (Atgs), including nuclear factor kappa B (NF-κB), hypoxia-inducible factor-1 (HIF-1), and Forkhead box O (FOXO) transcription factors. These factors orchestrate the expression of genes critical to autophagic flux in response to environmental stressors. Functionally, autophagy progresses through highly coordinated stages: initiation, vesicle nucleation, elongation, vesicle closure/retraction, and finally, fusion with lysosomes followed by cargo degradation. These steps are tightly regulated by a set of evolutionarily conserved proteins encoded by Atgs, ensuring precise control of autophagic activity in response to the cell’s physiological state [78].

The initiation phase of autophagy begins activating the Unc-51-like autophagy-activating kinase 1 (ULK1) complex, a crucial step in triggering the autophagic cascade. This complex comprises several essential components, most notably ULK1, focal adhesion kinase family-interacting protein of 200 kDa (FIP200), and autophagy-related protein 13 (Atg13). FIP200 and Atg13 play indispensable roles in stabilizing ULK1 and ensuring its proper localization to the sites where autophagy is initiated, specifically at the pre-autophagosomal structure (PAS) [79].

In conditions of nutrient deficiency or cellular energy stress, the AMP-activated protein kinase (AMPK), a central metabolic sensor, becomes activated. AMPK can phosphorylate the regulatory-associated protein of mTOR (Raptor) at specific serine residues (Ser722 and Ser792), thereby inhibiting the activity of the mechanistic/mammalian target of rapamycin (mTOR) signaling pathway. This inhibition prevents mTOR from phosphorylating ULK1 at its inhibitory site (Ser757), which is critical because phosphorylation of ULK1 at this site by mTOR complex 1 (mTORC1) suppresses autophagy initiation under nutrient-rich conditions [80].

When mTOR is inhibited and low energy levels, AMPK directly phosphorylates ULK1 at activating residues, leading to its full activation. This activation triggers the assembly and localization of the ULK1-Atg13-FIP200 complex at the PAS, which facilitates the recruitment of downstream autophagy machinery and promotes the nucleation of the phagophore, the initial sequestering membrane of autophagy [81].

By contrast, in nutrient-rich environments, mTORC1 remains active and phosphorylates ULK1 at Ser757, thus inhibiting its interaction with AMPK and suppressing the induction of autophagy [77]. This regulatory mechanism supports the role of nutrient availability in determining whether a cell initiates autophagy as a survival mechanism. Following the initiation step, the nucleation phase is characterized by the recruitment of lipids and proteins necessary to expand the autophagosome membrane. This step is mainly coordinated by the Beclin 1–class III phosphoinositide 3-kinase (PI3K) complex, which generates phosphatidylinositol 3-phosphate (PI3P). This lipid is a crucial player for membrane curvature and the recruitment of effector proteins promoting autophagosome formation [82]. This complex is central in defining the membrane origin and ensuring the spatial organization of new autophagosomal structures. The extension and maturation of the autophagosome membrane are primarily regulated by two autophagy-related proteins: Atg7 and Atg10. Among these, Atg7 plays a particularly critical role, serving as a functional mediator and as a molecular marker of autophagic vasculogenesis and membrane elongation. It orchestrates the formation of autophagic vesicles, often called autophagic “bubbles”. At this stage of autophagosome development, Atg7 catalyzes the conversion of the cytosolic form of the microtubule-associated protein light chain 3 (LC3I) into its lipidated form, LC3II—a process widely recognized as a defining hallmark of autophagy activation. The accumulation of LC3II on the autophagosome membrane is commonly used as a biomarker to assess autophagic activity.

Following vesicle expansion and elongation, the next critical step is the vesicle closure and maturation phase, during which the autophagosome fully forms. This process requires the recruitment and action of the Atg12–Atg5–Atg16L1 complex, which facilitates membrane curvature and autophagosome sealing. Subsequently, autophagolysosomes are generated through the fusion of mature autophagosomes with lysosomes a fusion event partly mediated by lysosomal-associated membrane protein 2 (LAMP2). These autophagolysosomes are responsible for the enzymatic degradation of intracellular waste, including damaged proteins, misfolded macromolecules, and dysfunctional organelles. The degradation products are then recycled as essential biomolecular precursors, which can be reused for biosynthesis or metabolized for energy production, marking the final phase known as the “fusion and degradation stage” [83]. Figure 2 provides a schematic overview of these molecular events governing the autophagy process.

However, several studies have reported that disruption or inhibition of the autophagosome–lysosome fusion step leads to a decrease in lysosomal degradation efficiency. This impairment can result in the accumulation of autophagosomes and a corresponding increase in the levels of LC3II and the LC3II/LC3I ratio. These findings suggest that elevated LC3II expression does not necessarily reflect effective autophagic degradation, as it may also indicate a blockage in downstream lysosomal processing [84].

Another key player in autophagy regulation is the autophagic adaptor protein p62, also known as “SQSTM1”. p62 mediates the selective degradation of protein aggregates by recognizing and binding to ubiquitinated substrates, guiding them toward the autophagic machinery. Notably, p62 levels are inversely correlated with autophagic flux. When autophagy is active and functional, p62 is efficiently degraded; however, when autophagy is impaired, p62 accumulates. Therefore, the degradation of p62 by autophagolysosomes is considered an indirect yet valuable indicator of lysosomal activity and autophagic efficiency [85].

In conclusion, it is now widely accepted that evaluating both the expression levels of p62 and the LC3 conversion rate (i.e., the LC3II/LC3I ratio) provides a more comprehensive and reliable assessment of the functional state of autophagy. These dual biomarkers help to distinguish between mere autophagosome accumulation and the actual progression of autophagic flux leading to substrate degradation.

### 2.5. Autophagy and Biological Systems

Emerging evidence underscores the critical involvement of autophagy in diverse biological systems, such as the cardiovascular, gastrointestinal, nervous, reproductive, and respiratory systems, where context-dependent modulation of autophagic activity profoundly influences physiological function and disease pathogenesis.

Autophagy plays a critical role in maintaining cardiovascular homeostasis by removing damaged organelles and misfolded proteins, thus preserving cardiomyocyte function and survival. In the heart, basal autophagy is essential for normal physiology, while dysregulation contributes to various pathologies, including ischemic heart disease, hypertrophy, and heart failure [86]. During ischemia-reperfusion injury, autophagy is initially protective by eliminating damaged mitochondria (mitophagy), but excessive activation may lead to cell death, suggesting a biphasic role [87]. Additionally, endothelial autophagy regulates vascular integrity and inflammation by modulating nitric oxide bioavailability and suppressing oxidative stress [88]. Emerging evidence also implicates autophagy in the pathogenesis of atherosclerosis through macrophage lipid handling and plaque stability [89]. Therapeutic modulation of autophagy, such as via AMPK or mTOR signaling, is being explored to mitigate cardiovascular disease progression [90]. However, a nuanced understanding of temporal and tissue-specific autophagic responses is essential to avoid detrimental outcomes.

In the intestinal epithelium, autophagy plays a pivotal role in maintaining mucosal integrity, regulating immune responses, and protecting against oxidative stress. The various intestinal epithelial cells, including enterocytes, Paneth cells, and goblet cells, rely on autophagy to modulate the secretion of antimicrobial peptides and mucins, essential for host defense and microbiota regulation. Autophagy in Paneth cells is vital for the secretion of defensins, while goblet cells regulate mucin production, contributing to the protective function of the intestinal mucosa [91]. Dysregulation of autophagic mechanisms in the intestinal epithelium has been linked to several intestinal disorders. Impaired autophagy in epithelial cells has been shown to disrupt the intestinal barrier, increasing permeability and contributing to chronic inflammation, a hallmark of inflammatory bowel diseases (IBD) such as Crohn’s disease and ulcerative colitis [92]. Genetic variants in autophagy-related genes such as ATG16L1 and immunity-related GTPase family M protein (IRGM) have been implicated as susceptibility factors in the development of IBD, underscoring the importance of autophagy in maintaining intestinal immune homeostasis [93]. Additionally, autophagy plays a critical role in modulating oxidative stress within intestinal epithelial cells. It limits the production of reactive oxygen species (ROS) and regulates the activation of the NOD-like receptor family pyrin domain-containing 3 (NLRP3) inflammasome, both of which are involved in the pathogenesis of IBD [94]. These findings suggest that autophagy modulation could be a therapeutic approach to restore mucosal homeostasis and alleviate chronic inflammation in conditions like IBD.

Autophagy is indispensable for neuronal health, given neurons’ high metabolic demands and post-mitotic nature. It facilitates the clearance of damaged mitochondria and aggregated proteins, thereby preserving synaptic function and neuroplasticity [95]. Impaired autophagy has been implicated in the pathogenesis of numerous neurodegenerative disorders, including Alzheimer’s disease, Parkinson’s disease, and amyotrophic lateral sclerosis (ALS) [96]. In Alzheimer’s disease, accumulation of autophagic vacuoles and defective lysosomal clearance contribute to β-amyloid and tau pathology [97]. Similarly, defective mitophagy leads to dopaminergic neuron degeneration in Parkinson’s disease via the accumulation of dysfunctional mitochondria [98]. Autophagy also protects glial cells, modulating neuroinflammation and blood–brain barrier integrity [99]. Significantly, autophagy intersects key signaling pathways such as mTOR, AMPK, and transcription factor EB (TFEB), making it a viable target for therapeutic intervention. Enhancing autophagy pharmacologically or genetically has shown promise in ameliorating neurodegenerative phenotypes in preclinical models.

Autophagy regulates essential processes in both male and female reproductive systems, including gametogenesis, steroidogenesis, and embryonic development. In ovarian physiology, autophagy modulates follicular atresia and oocyte quality by mediating organelle turnover and energy homeostasis [100]. Dysregulated autophagy in granulosa cells is linked to polycystic ovary syndrome (PCOS) and ovarian aging [101]. In male reproduction, autophagy supports spermatogenesis by eliminating residual cytoplasm and ensuring mitochondrial quality control, particularly during spermiogenesis. Sertoli cell-specific deletion of Atg7 impairs spermatogenesis and leads to male infertility, underlining its critical function [102]. Moreover, autophagy intersects with hormonal signaling; for example, luteinizing hormone (LH) and follicle-stimulating hormone (FSH) modulate autophagic flux in reproductive tissues [103]. During early embryogenesis, autophagy ensures cellular remodeling post-fertilization and is vital for preimplantation development [104]. While autophagy appears protective in most reproductive contexts, excessive or defective autophagic activity may contribute to infertility, endometriosis, and reproductive senescence, suggesting a therapeutic window for modulation.

In the context of the respiratory system, autophagy plays a pivotal role in maintaining the structural and functional integrity of the airway epithelium, an essential barrier and immunological interface that lines the conductive airways. The airway epithelium serves as the first line of defense against environmental insults, including airborne pathogens, allergens, and particulate matter. It comprises a heterogeneous population of ciliated, secretory, and basal cells that undergo constant renewal and are highly responsive to physiological and pathological stimuli. Autophagy in these epithelial cells contributes to tissue homeostasis through several mechanisms: regulating cellular differentiation and turnover, preserving mitochondrial quality, modulating inflammatory responses, and maintaining mucociliary clearance. Under basal conditions, autophagic flux in airway epithelial cells supports the degradation of dysfunctional mitochondria (mitophagy) and the turnover of cytoplasmic debris, ensuring optimal cellular function. During exposure to respiratory pathogens such as the influenza virus, the respiratory syncytial virus (RSV), and *Pseudomonas aeruginosa*, autophagy mediates pathogen clearance (xenophagy) and restricts viral replication. Moreover, autophagy intersects with pattern recognition receptor (PRR) signaling pathways to fine-tune the innate immune response, limiting excessive inflammation and tissue injury. Impaired autophagic activity in the airway epithelium has been implicated in the pathogenesis of several chronic pulmonary diseases. Defective autophagy has been associated with goblet cell hyperplasia, excessive mucus production, and impaired epithelial repair in asthma. In chronic obstructive pulmonary disease (COPD), dysregulated autophagy contributes to epithelial cell senescence, oxidative stress, and impaired barrier integrity, promoting airway remodeling and progressive airflow limitation. Notably, cigarette smoke—a primary etiological factor in COPD—has been shown to induce aberrant autophagic responses that exacerbate epithelial injury and inflammation [105]. Recent evidence also highlights the role of autophagy in regulating epithelial responses to oxidative stress and environmental pollutants such as ozone and diesel exhaust particles. By modulating ROS levels and influencing redox-sensitive signaling cascades, autophagy confers cytoprotection and mitigates epithelial cell apoptosis. Additionally, autophagy supports ciliogenesis and the maintenance of ciliary function, which are essential for adequate mucociliary clearance and airway defense [106].

These findings suggest that modulating autophagy could offer therapeutic benefits across various organs and pathological contexts.

### 2.6. Nanoplastics and Autophagy

To better understand how NPs affect autophagy, we searched PubMed using the keywords “autophagy” and “nanoplastic”. We selected studies that used both in vitro and in vivo models and grouped them by organ system. Most of these studies focused on PS-NPs, which are often used in toxicity research. The results show that NPs can interfere with autophagy in many tissues. However, the effects and mechanisms change depending on the organ studied. Autophagy is a key cellular process. It removes damaged organelles and proteins, helping the cell maintain balance and health. However, when autophagy does not work correctly, it can lead to diseases like cancer, neurodegeneration, and metabolic problems. As shown in Table 1, most of the current research focuses on PS. PS is the main plastic studied in all models, from the heart to the brain. In comparison, few studies use other polymers, such as PET, PMMA, PVC, PP, PE, and PTFE. This creates a narrow view. Relying only on PS does not reflect real environmental conditions. PS is common, but not the only one we are exposed to. For example, PE is everywhere. It is used to make containers, bottles, tubing, plastic bags, and lab tools. PE is one of the most produced plastics in the world and the most found in marine litter [107,108]. Its structure, with -CH_2_ groups, makes it hydrophobic and able to absorb pollutants [109]. Another example is PVC, the third most used synthetic plastic. It is thermoplastic so that it can be reshaped many times. Its polar nature allows for mixing with many additives. It is used in pipes, cables, packaging, textiles, and medical devices [110,111]. The lack of polymer variety in research limits our understanding. We still do not know how different plastics affect biological systems, especially regarding autophagy. As Table 1 shows, PS is present in almost every organ study.

#### 2.6.1. Cardiovascular System

In cardiovascular models, PS-NPs caused oxidative stress, mitochondrial abnormalities in terms of morphology, and blocked proper autophagy. Autophagy was activated in hESCs, Zebrafish, and H9C2 heart cells, but it did not always protect them. Sometimes, it seemed to cause damage instead. Autophagy happens in steps: first, autophagosomes form, then they fuse with lysosomes, and finally, the content gets degraded. Li et al. (2024) reported that PS-NPs increased early markers like LC3-II/I and Atg5, showing autophagosomes were being made. However, levels of p62 went up, and LAMP1 went down, meaning the later steps, fusion and degradation were not working well. So even though autophagy started, it could not finish (Figure 3). The whole process got stuck [112]. Many studies now link this kind of defective autophagy to heart disease. It is a key system for keeping heart cells balanced. Peng et al. (2022) showed that PS-NPs, alone or with lipopolysaccharide (LPS), triggered oxidative stress and activated the ROS/TGF-β1/Smad pathway in mouse hearts and H9C2 cells. This led to both fibrosis and autophagy. They also activated the AMPK/mTOR/ULK1 pathway, raising Beclin1 and LC3-II levels and lowering p62. With LPS, the damage got worse. Here, autophagy was not helpful. TGF-β1 was a key factor driving both fibrosis and autophagy. When they blocked it using LY2109761, some effects were reversed. However, overall, PS-NPs worsened LPS-induced heart injury through autophagy dysfunction, driven by oxidative stress and TGF-β1 signaling [113].

As reported in Section 2.3, sizes play a crucial role in internalization and toxicity, but this concept is also related to autophagy. Yan-Yang Lu et al.’s (2022) study showed that particle size influences how PS-NPs interact with HUVECs. When they were exposed to 500 nm PS-NPs, almost all particles attached to the cell membrane but did not enter the cells. This strong membrane binding caused visible damage and increased the release of lactate dehydrogenase (LDH). However, no autophagy activity was detected in this condition. In contrast, 100 nm PS-NPs were internalized into the cytoplasm. These smaller particles induced the formation of autophagosomes and increased the expression of autophagy-related proteins such as LC3B-II and BECN1. However, they also blocked autophagic flux, likely due to impaired lysosomal function. An apparent reduction in LAMP2 and Cathepsin B (CTSB) levels confirmed this. These results suggest that only smaller NPs (100 nm) can trigger and disrupt autophagy in HUVECs [114].

#### 2.6.2. Gastrointestinal System

The gastrointestinal (GI) system is one of the most studied organs regarding autophagy triggered by NPs, as shown in Table 1. In this context, particle size seems to play a crucial role. Smaller particles can more easily compromise the intestinal barrier, especially affecting the tight junctions (TJs). GI epithelial cells are the first barrier that protects the digestive tract. In contrast, the TJs serve as a second layer of defense that stops harmful substances like antigens, bacteria, or toxins from entering the bloodstream. Recent studies suggest that autophagy may help maintain the integrity of this barrier. For example, Saha et al. (2023) found that autophagy supports the membrane localization of occludin, a key TJ protein. This process can help strengthen the barrier and reduce damage caused by inflammation [115].

A review by Chang et al. (2024) in the *World Journal of Gastroenterology* showed that autophagy plays a significant role in the function of different GI cells. This includes digestive, secretory, and regenerative cells. The review also explained that autophagy can help regulate inflammation, essential for keeping the gut balanced [116]. It is known that many nanoparticles can induce autophagy, which sometimes leads to toxicity [117]. Xu et al. (2023) showed that PS-NPs (100 nm) triggered autophagy in human intestinal cell lines (HIEC-6, RKO, HT-29, HCT-116) and mice. Increased LC3-II and p62 levels confirmed this. However, autophagic flux was blocked. There was lysosomal accumulation and autophagosome buildup. In mice, the colon tissue did not show significant visible damage [118].

Also, NPs affect the environment, and this is important to improve awareness of this new emerging pollutants. Li et al. (2024) found that NPs ingested by the hepatopancreas damaged tissue and increased cell death and autophagy-related structures. This response was linked to oxidative stress, activating lysosomes, DNA repair, and immune pathways. All this may lead to inflammation, autophagy, and apoptosis [119].

Regarding the esophagus, autophagy—especially mitophagy—is a key process for removing damaged mitochondria [120,121]. A study by Guanglin et al. (2024) found that exposure to PS-NPs resulted in a reduction in mitophagy (Table 2). The problem seemed to be linked to a ferrous ion (Fe^2+^) overload. However, the exact molecular pathway is still unclear and needs further research. Furthermore, the same study found that PS-NPs caused inflammation and cell death in esophageal cells (HET-1A and HEEC). Mechanistic analysis revealed that PS-NPs increased levels of Fe^2+^, leading to an overproduction of ROS (Figure 3). This resulted in cell damage and inflammation. PS-NPs also appeared to block mitophagy, exacerbating the inflammation caused by Fe^2+^ overload. These results enhance our understanding of how PS-NPs can harm the esophagus [122]. As reported in Table 2, current studies primarily focus on PS, while other polymer autophagy–behavioral are still not studied.

#### 2.6.3. Nervous System

Neurons are highly polarized cells with a large amount of cytoplasm. Consequently, they can easily suffer if cell waste builds up. Therefore, neurons are particularly susceptible to problems in the autophagy process. Autophagy maintains cellular stability and keeps neurons stable by removing damaged proteins and preventing brain disorders [123]. This process is also important for brain activity, plasticity, and memory [124]. Not only neurons are affected—autophagy problems also happen in microglia [125], and astrocytes [99]. More studies now demonstrate that abnormal autophagy is linked to many brain diseases, like Alzheimer’s [126], Huntington’s [127], and Parkinson’s [128]. PS-NPs have been shown to cross the BBB, damaging brain function and changing behavior in mice [75,129].

Mitochondria are the primary source of energy in the cell and play a role in many diseases, especially brain diseases like Parkinson’s [130]. In this disease, mitochondria do not work well—the electron chain is weak, their shape changes, DNA can be damaged, and calcium balance is disturbed [131]. How PS-NPs affect mitochondria in dopaminergic neurons still needs more research. Mitophagy is a special kind of autophagy that takes care of mitochondria. It helps maintain stable energy levels and healthy neurons [132]. However, when mitophagy is excessive, either too much or too little, it harms the cell [121,133].

In 2023, Huang et al. demonstrated that PS-NPs can cause too much mitophagy. In SH-SY5Y cells, PS-NPs damaged mitochondria by reducing complex I activity of the electron transport chain and ATP levels, and they activated the AMPK/ULK1 pathway. This turned on the PTEN-induced kinase 1 (PINK1)/Parkin pathway too strongly, leading to neuronal damage and cell death (Figure 3). The same pathway was activated in mice, resulting in the loss of dopaminergic neurons and movement disorder [134].

Autophagy is also involved in the health of brain blood vessels. It regulates TJ proteins, which maintain the strength of BBB. When autophagy is not well balanced—either too much or too little—it can damage these TJ proteins and weaken the BBB [135,136].

Iron levels inside cells also affect the BBB. In particular, Fe^2+^ can increase oxidative stress, damage lipids, and disturb cell balance. The autophagy pathway controls iron recycling through the lysosomes [137]. As iron builds up, it can lead to ferroptosis, a kind of cell death dependent on autophagy [138]. This process results in increased permeability and damage of the BBB [139]. To support this, Kim E et al. (2025) investigated the effect of NPs on the BBB using brain endothelial cells. This iron overload, combined with inhibited autophagy, increased BBB permeability and damaged the barrier in treated bEnd.3 cells [140].

#### 2.6.4. Reproductive System

Autophagy is a key process that keeps cells clean and promotes their survival. It also plays a significant role in the reproductive system. In males, autophagy supports spermatogenesis and maintains the health of Sertoli and Leydig cells. If autophagy is not functioning correctly, it can decline sperm quality and function [141]. In females, autophagy helps in the development of oocytes and the survival of follicles. It protects the ovaries from oxidative stress and keeps eggs healthier, especially as women age [142]. Thus, normal autophagy is important for fertility in both men and women.

Recent studies have shown that PS-NPs can reach offspring after a mother is exposed to them. For example, in a model using isolated human placentas, PS-NPs crossed the placental barrier and reached the placenta [143]. In animal studies, maternal exposure to PS-NPs by intratracheal drip led to their accumulation in the placenta and fetus. It was also observed that NPs can pass through the placenta and breast milk, reaching the pups and affecting their development [144].

In a recent study, Ma et al. (2025) discovered that maternal exposure to PS-NPs reduced autophagy in undifferentiated spermatogonia in male offspring. They noticed lower autophagic activity, damaged mitochondria, and unusual lysosomes. The levels of p62 protein increased while the LC3-II/I ratio decreased, which means autophagy was blocked. Some important autophagy genes were also reduced, like Tcirg1, Igbp1, and Gabarapl2. These effects were linked to a higher expression of Prdm14, a transcription factor activated by inflammation. This shows that PS-NPs might change autophagy in a harmful way and cause poor sperm development in male pups [145].

Compared to MPs, NPs are smaller and have a larger surface area. They can enter cells more easily and interact more with biological systems. As a result, their toxicity is more substantial, and they can carry more other pollutants [146]. Today, many studies focus on how NPs and other pollutants interact to affect the female reproductive system, primarily through autophagy. Triclosan (TCS) is one of these pollutants. It is an antibacterial substance in everyday products like toothpaste, soap, mouthwash, and cosmetics [147]. TCS can damage the ovaries by causing oxidative stress, harming mitochondria, disrupting lipid balance, and triggering apoptosis [148]. PS-NPs can adsorb chemicals like TCS very well. Their binding is strong and stable [149].

In a recent study, Wang et al. (2024) tested the combined effect of PS-NPs and TCS on human granulosa cells. They found that co-exposure caused high cytotoxicity. There was increased oxidative stress, mitochondrial damage, and a strong activation of autophagy. The LC3B-II/LC3B-I ratio and ATG5 levels increased while p62 protein decreased, which indicates that the autophagic flux was active and not blocked. However, this overactivation was not helpful—it was linked to more cell death. These results suggest that PS-NPs can harm female reproductive cells by overstimulating oxidative stress and autophagy, especially when mixed with common additives like TCS, as shown in Figure 3 [150].

Recent evidence has revealed that NPs exposure may modulate autophagic flux through epigenetic interference. Shen et al. (2025) investigated the effects of chronic exposure to 0.1 μm PS microplastics in male mice and GC-1 spermatogonia cells, uncovering that PS accumulation disrupts both epigenetic regulation and autophagy-related pathways. Notably, they observed a downregulation of Histone H3 lysine 9 dimethylation (H3K9me2), a key histone modification associated with gene repression, coupled with increased expression of Lysine-specific demethylase 3A (KDM3A), a demethylase involved in transcriptional activation. This epigenetic imbalance correlated with enhanced expression of glycolytic enzymes such as Pyruvate Kinase M2 (PKM2), Lactate Dehydrogenase A (LDHA), and Glyceraldehyde-3-phosphate dehydrogenase (GAPDH). At the same time, autophagic markers LC3B and p62 were elevated, indicating a disruption in autophagy flux. Co-immunoprecipitation studies further revealed an interaction between p62 and PKM2, suggesting impaired autophagic degradation of glycolytic enzymes. The study proposes that dysfunctional autophagy, driven by NPs accumulation, exacerbates the accumulation of metabolic enzymes due to epigenetic deregulation, leading to defective spermatogenic differentiation and reduced fertility. These findings support the concept that NPs interfere with cell homeostasis by targeting the intricate crosstalk between metabolism, epigenetic state, and autophagic responses [151].

#### 2.6.5. Respiratory System

Lung inflammation happens when the respiratory tract is often exposed to microbes, air particles, pollutants, irritants, allergens, or pathogens [152]. Autophagy plays a key role in how the lung responds to stress and infection [153]. Studies in mice show that when autophagy does not work correctly, they develop lung inflammation even without infection. This condition is characterized by signs such as inflammatory cells, thicker tissue under the surface, goblet cell changes, and more collagen in the lungs [154]. Because of this, it is crucial to understand how NPs affect autophagy. Annangi et al. (2023) found that PS-NPs inside cells lead to high levels of ROS, loss of mitochondrial membrane potential (MMP), and alterations in autophagy (Figure 3). One significant result was that autophagy was blocked. This could help explain other harmful effects of PS-NPs on cells. When autophagy does not function properly, it may even support cell transformation. This could be dangerous, especially now that people are exposed to more PS-NPs from the environment [155].

In the lungs, BEAS-2B and nasal epithelial cells exhibited signs of ferroptosis and ferritinophagy after PS exposure. A research paper reported that PS damaged mitochondria and harmed lung cells by altering autophagy [156]. ROS levels and LC3-II increased, indicating that autophagy was affected. Still, most of these studies only used polystyrene. There is very little data on other plastics. One study did test PET on HNEpCs, showing more LC3-II and p62 proteins [157]. However, this is just one example among many studies focused on PS.

**Table 2 ijms-26-07035-t002:** Cellular and systemic effects of nanoplastics in experimental models.

	Model				
Nanoparticles	In Vitro	In Vivo	Effects	System	References
PS	hESCs	*Zebrafish*	Cytotoxicity, affected differentiation, oxidative stress and mitochondrial dysfunction, disturbed autophagy flux, impaired cardiac function.	Cardiovascular	[112]
PS	H9C2	mouse	Oxidative stress, activation of TGF-β1 and autophagy.	Cardiovascular	[113]
PS	HUVECs		Cell membrane damage, inducing autophagosome formation and blocking autophagic flux.	Cardiovascular	[114]
PS	CACO-2	Forty male-ICR mice	Intestinal barrier damage and enterocyte apoptosis, lysosomal dysfunction and autophagic substrate degradation arrest in enterocytes.	GI	[158]
PS		*L. vannamei*	Destruction of hepatopancreas tissue structure, the shedding of microvilli, the increase number of hepatocyte apoptosis and autophagy structure.	GI	[119]
PS	RKO HT-29 HCT-116 HIEC-6	C57BL/6J mice	Induced autophagy activation, but blocked autophagic flux in human intestinal cells. No major tissue damage observed in vivo.	GI	[118]
PS		*M. nipponense*	Inducing apoptosis and autophagy.	GI	[159]
PS		*L. vannamei*	Damaging of the intestinal villi, promotion and formation of autophagosomes, increasing of intestinal non-specific immunoenzyme activities, and significantly induction of apoptosis.	GI	[119]
PS	HET-1A HEET		Suppressing mitochondrial autophagy, which exacerbated NP-induced cell inflammation and death.	GI	[122]
PS	HIEC-6		Inhibition of mitophagy and inducing perturbations in the gut microbiota.	GI	[160]
PS	GES-1		Decreasing cell proliferation rates and increasing cell apoptosis and autophagy flux.	GI	[161]
PS	SH-SY5Y	C57BL/6 J mice	Activation of the AMPK/ULK1 pathway driving excessive mitochondrial autophagy, death of dopaminergic neuron.	Nervous	[134]
PS		Sprague Dawley rats	Induction of autophagy.	Nervous	[162]
PS-NH2 (modified)	bEnd.3		Inhibition of autophagy pathway in brain endothelial cells, decreased mTOR phosphorylation, altered Beclin-1 and LC3B ratios, and p62 accumulation which contributed to iron overload and subsequent blood–brain barrier disruption.	Nervous	[140]
PS	Swan 71	C57BL/6 mice	Suppression of ROCK1-mediated migration/invasion and migrasome formation. Activation of autophagy and promotion of the autophagy degradation of SOX2, thus suppressing SOX2-mediated ROCK1 transcription. Damage to trophoblast cells and placenta tissue and induction of miscarriage.	Reproductive	[163]
PS	GC-2spd(ts)		Inhibition of cell proliferation and decreasing of cell viability, induction of oxidative stress and autophagy, impairing mitochondrial function of spermatocyte.	Reproductive	[164]
PS		Kunming mice	PS-NPs reduced autophagic flux in offspring undifferentiated spermatogonia, as demonstrated by decreased expression of key markers such as Igbp1/Gabarapl2 and structural disruptions in autophagolysosome.	Reproductive	[145]
PS	KGN		Triggered autophagy, increased LC3B-II/LC3B-I ratio and elevated ATG5 expression, coupled with a reduction in P62 levels.	Reproductive	[150]
PS	BEAS-2B HPAEpiC		Inducing lung epithelial cell ferroptosis and ferritinophagy, disturbing of mitochondrial functions and damage, triggering autophagy.	Respiratory	[156]
PS	HNEPCs		Significant increases in ROS, a decrease in MMP (mitochondrial membrane potential), as well as a greater accumulation of LC3-II and p62.	Respiratory	[155]
PS	HeLa		Activating of autophagic flux.	Respiratory	[165]
PS	BEAS-2B		Inducing of oxidative stress and inhibited cell growth through apoptosis and autophagy.	Respiratory	[166]
PET	HNEpCs		Induction of ROS’s production and alteration of autophagy when LC3-II and p62 protein’s levels increase.	Respiratory	[157]
PS	BEAS-2B		Metabolic alterations related to autophagy and endoplasmic reticulum (ER) stress, such as an increase in amino acids and tricarboxylic acid (TCA) cycle intermediate metabolites.	Respiratory	[167]

### 2.7. Molecular Signaling Pathways Linking NPs to Autophagy Dysregulation

Recent findings have elucidated the intricate molecular mechanisms through which NPs, particularly PS-NPs, modulate autophagy, either as a protective cellular response or as a contributor to toxicity, depending on the context and degree of exposure. Among the earliest events triggered by NPs is the generation of reactive oxygen species [168], which acts as a central upstream signal initiating autophagic responses. ROS accumulation leads to the activation of AMP-activated protein kinase (AMPK), a master regulator of cellular energy homeostasis and an established positive modulator of autophagy. Activated AMPK phosphorylates Unc-51-like autophagy-activating kinase 1 (ULK1) at Ser317 and Ser777, thereby promoting autophagy initiation. Concomitantly, AMPK inhibits mechanistic target of rapamycin complex 1 (mTORC1) by phosphorylating Raptor at Ser722 and Ser792, relieving the inhibitory phosphorylation of ULK1 at Ser757 and further enhancing autophagy initiation [80,81]. This AMPK–mTOR–ULK1 axis is a recurring motif in NP-induced autophagy across various biological systems. For instance, in SH-SY5Y neuronal cells, PS-NPs induced oxidative stress and activated AMPK, resulting in robust ULK1 phosphorylation and mitophagy via the PINK1/Parkin pathway, ultimately leading to dopaminergic neuronal injury and impaired motor function in vivo [134]. A similar pattern has been documented in cardiovascular cells, where PS-NPs or PS-NPs in combination with lipopolysaccharide (LPS) exposure activated the ROS/TGF-β1/Smad signaling cascade and engaged the AMPK–mTOR–ULK1 axis, enhancing autophagosome formation while compromising flux, as evidenced by elevated Beclin1 and LC3-II levels coupled with p62 accumulation [113]. Disruption of autophagic flux appears to be a defining feature of NP exposure, particularly due to impaired lysosomal function. Studies in intestinal and endothelial cell lines have reported decreased expression of LAMP1, LAMP2, and CTSB following NP exposure, indicating a blockage in autophagosome–lysosome fusion [112,114,118]. The persistent elevation of autophagy markers such as LC3-II and p62 in these contexts underscores a scenario of stalled autophagy rather than enhanced flux. Furthermore, NP-induced oxidative stress has been shown to stabilize transcription factors such as FOXO and HIF-1α, which are known to upregulate autophagy-related genes (e.g., Atg5, BECN1) at the transcriptional level, potentially enhancing the early stages of autophagy while failing to complete the degradative process due to lysosomal insufficiency [78,80].

Molecular cross-talk between autophagy and inflammatory signaling is also evident in NP toxicity. Transforming growth factor beta 1 (TGF-β1), a key mediator of fibrosis and inflammation, was found to be upregulated in NP-exposed cardiac cells, where it contributed to Smad2/3 phosphorylation and fibrotic remodeling in tandem with dysregulated autophagy. Pharmacological blockade of TGF-β1 partially restored autophagic flux and reduced tissue injury, indicating its pivotal role in linking NP-induced inflammation and autophagic imbalance [113]. Finally, ferroptosis-related signaling is tightly linked with NP-altered autophagy in respiratory models. PS-NPs increased intracellular iron (Fe^2+^), ROS, and lipid peroxidation, concurrently impairing autophagy and promoting ferroptotic cell death. This was reflected by increased LC3-II and p62 levels and altered Beclin1 expression in brain and lung endothelial cells [140,156,157]. Since ferritinophagy—a form of selective autophagy—regulates intracellular iron, its dysfunction under NP exposure may exacerbate oxidative injury and barrier disruption in critical tissues such as the blood–brain barrier and airway epithelium [137,138,139]. NPs manipulate autophagy via a multilayered network of molecular signaling pathways, including but not limited to AMPK–mTOR–ULK1, PINK1/Parkin, ROS–TGF-β1–Smad, and epigenetic modifiers. The precise outcome—whether protective or deleterious—appears to depend on particle characteristics, cell type, exposure duration, and the integrity of downstream autophagic machinery. A more nuanced understanding of these pathways is essential for developing targeted interventions to mitigate NP-induced toxicity in human health and ecosystems.

## 3. Conclusions

This review highlights that NPs, particularly PS-NPs, can significantly interfere with autophagy, the cell’s cleaning system involved in degrading and recovering damaged components. The gastrointestinal tract, brain, lungs, cardiovascular, and reproductive systems appear particularly vulnerable among organ systems. The most frequently encountered effects are altered autophagy flux, mitochondrial dysfunction, and increased oxidative stress.

Despite the growing scientific interest in this field, current evidence appears limited and fragmentary. First, available data are mainly derived from studies focusing on PS-NPs. However, NP exposure also affects other plastic polymers, including PE, PP, PMMA, PVC, and PET, whose effects in terms of modulating the autophagic process are still largely unexplored. Distribution of size, polymer type, surface properties, and biocorona composition are just some of the factors that potentially contribute to the differential modulation of the autophagy process in various organ systems.

Second, in many studies, autophagy has been studied indirectly, often relying on the evaluation of single markers. In addition, studies conducted with in vitro cell models often cannot capture the dynamic and multistep nature of the autophagy process. These limitations hinder elucidation of the mechanisms by which NPs modulate autophagy in different cells and tissues. These knowledge gaps suggest the urgent need for more comprehensive and mechanism-oriented research approaches.

Current evidence on NP-induced modulation of autophagy in immune system cells remains very limited. Given the central role of autophagy in regulating immune cell function, inflammation, and immune homeostasis, further studies are needed to clarify how different types of NPs may impact autophagic pathways in various immune cell populations. Addressing this gap could provide important insights into the immunotoxic potential of NPs and their broader implications for human health.

A broader consideration is that NP toxicity studies do not provide exposure-response information suitable for risk management. In the future, the study of the toxic effects of NPs should broaden the range of NP types studied and include long-term, low-dose exposure models that better reflect real-world environmental conditions. In addition, research should address the combined effects of NPs with other environmental pollutants, as co-exposures will likely represent human and ecological reality more.

## Figures and Tables

**Figure 1 ijms-26-07035-f001:**
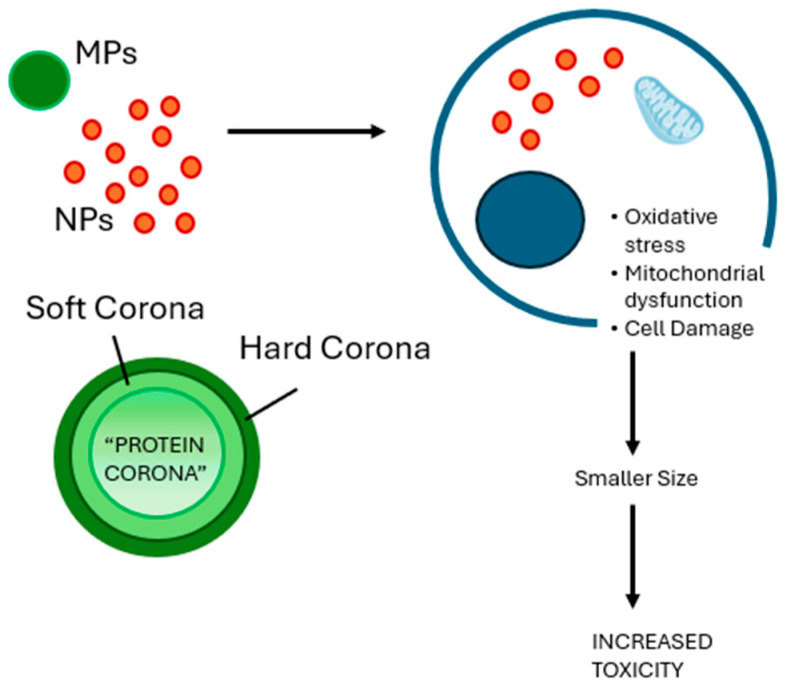
**Representative illustration of the protein corona and cell damage induced by MPs and NPs.** Overview of major cell damage induced by MPs and NPs, and schematic representation of the protein corona, with distinction in the hard and soft corona. Smaller size plays a crucial role in terms of toxicity. MPs: microplastics; NPs: nanoplastics.

**Figure 2 ijms-26-07035-f002:**
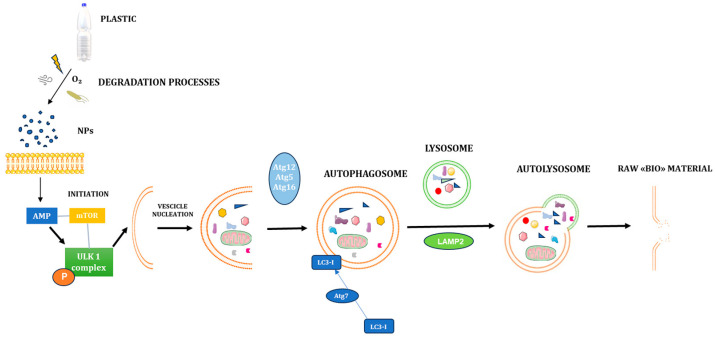
**Schematic representation of autophagy mechanism.** Plastics are durable, low-cost synthetic polymers used widely, but once in the environment, they degrade physically, chemically, and biologically. Factors like UV light, abrasion, microbes, and oxidation break plastics into smaller pieces, eventually forming nanoplastics (NPs). These NPs, due to their tiny size and large surface area, are more reactive and can more easily interact with biological membranes. NPs can cross cellular barriers, be internalized by endocytosis or passive diffusion, and accumulate within cells. This internalization may trigger cellular stress responses, as NPs can carry chemicals, heavy metals, and other pollutant agents on their surface. Moreover, NPs have a more reactive surface, which may generate reactive oxygen species (ROS) and induce oxidative stress, including the activation of the autophagy pathway. Autophagy is a lysosomal-mediated degradation process that primarily degrades damaged cells and dysfunctional organelles to recycle damaged or dysfunctional cellular contents, thereby providing the energy for nascent cells. The initiation of autophagy is regulated by two main sensors: AMPK and mTOR. mTOR and AMPK in the initial steps of the autophagy process through phosphorylation interaction with the ULK1 complex, respectively, whereas AMPK inhibits mTOR activity. AMPK: AMP-activated protein kinase; Atg: autophagy-related gene; LC3: microtubule-associated protein 1A/1B-light chain 3; mTOR: mammalian target of rapamycin; P: phosphate; ROS: reactive oxygen species; ULK1: Unc-51 like autophagy; NPs: nanoplastics; UV: ultraviolet. Created with Bioicons (https://bioicons.com/, accessed on 5 May 2025), licensed under CC BY 4.0.

**Figure 3 ijms-26-07035-f003:**
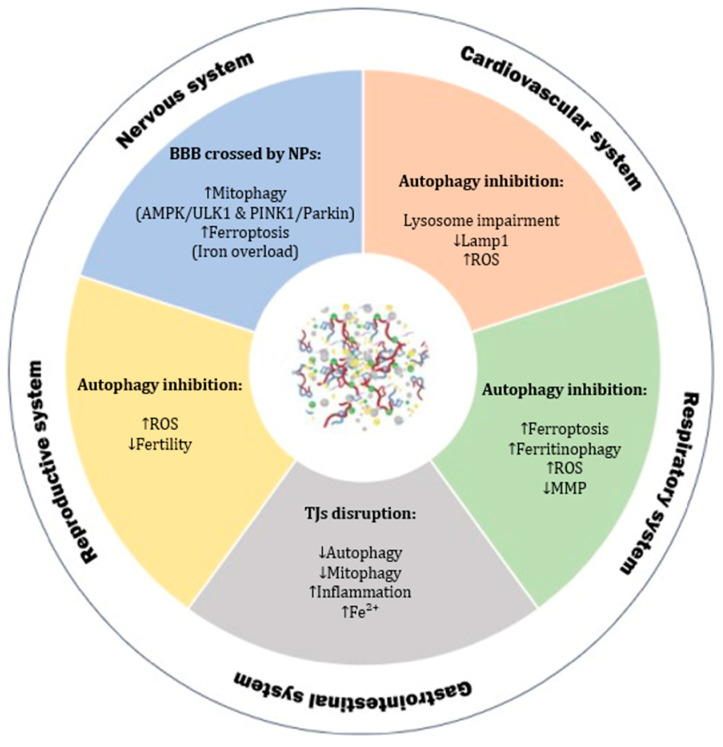
**Overview of the systemic effects of NPs and their impact on autophagy in different organs.** NPs can cross biological barriers and accumulate in various tissues, interfering with autophagy and related pathways. In the nervous system, NPs cross the BBB, promoting mitophagy (via AMPK/ULK1 and PINK1/Parkin pathways) and ferroptosis due to iron overload. In the cardiovascular system, NPs inhibit autophagy through lysosomal impairment, decreased LAMP1 expression, and increased ROS. In the respiratory system, they alter mitochondrial function, enhance ferroptosis and ferritinophagy, and reduce MMP. In the gastrointestinal system, NPs disrupt TJs, reduce autophagy and mitophagy, and trigger inflammation and iron dysregulation. Finally, NPs lead to autophagy inhibition in the reproductive system, increased ROS production, and reduced fertility. NPs: nanoplastics; BBB: blood–brain barrier; AMPK: AMP activated protein kinase; ULK1: Unc-51 like autophagy; PINK1: PTEN induced kinase 1; LAMP1: lysosomal-associated membrane protein 1; ROS: reactive oxygen species; MMP: mitochondrial membrane potential; TJs: tight junctions.

## Data Availability

No new data were created or analyzed in this study.

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
