# Peer review of "Biological Modulation of Autophagy by Nanoplastics: A Current Overview"

_ijms, 2025, doi:10.3390/ijms26157035_

Round 1

Reviewer 1 Report

Comments and Suggestions for Authors

The authors that summarized the recently data on nanoplastics (NPs) that an emerging class of environmental pollutants, are increasingly recognized for their potential to interfere with critical cellular processes. They found that nanoplastics (NPs) is existing in whole body, including the gastrointestinal tract, brain, lungs, cardiovascular, and reproductive systems. More important can affect the autophagy process, mitochondrial dysfunction, and increased oxidative stress in vitro and in vivo. They are point out the NP–autophagy crosstalk for evaluating NP toxicity and guiding future research in environmental health and nanotoxicology. It is interesting. Some comments as following during publish.

  1. In Figure 2, in the figure legend: NPs can cross cellular barriers, be internalized by endocytosis, and accumulate within cells. This internalization may trigger cellular stress responses, including the activation of the autophagy pathway. If it is possible, said it for more clear is how that NP internalization may trigger cellular stress responses and autophagy flux.

  1. Please check it whether can affect the immune system such as T cell, B cell and DC cells?

  1. And please checked it how about in epigenetic effect?

Author Response

Reviewer #1 (Technical Comments to the Author): 

The authors that summarized the recently data on nanoplastics (NPs) that an emerging class of environmental pollutants, are increasingly recognized for their potential to interfere with critical cellular processes. They found that nanoplastics (NPs) is existing in whole body, including the gastrointestinal tract, brain, lungs, cardiovascular, and reproductive systems. More important can affect the autophagy process, mitochondrial dysfunction, and increased oxidative stress in vitro and in vivo. They are point out the NP–autophagy crosstalk for evaluating NP toxicity and guiding future research in environmental health and nanotoxicology. It is interesting. Some comments as following during publish.

  1. In Figure 2, in the figure legend: NPs can cross cellular barriers, be internalized by endocytosis, and accumulate within cells. This internalization may trigger cellular stress responses, including the activation of the autophagy pathway. If it is possible, said it for more clear is how that NP internalization may trigger cellular stress responses and autophagy flux.

Thank you for your valuable suggestion. We have revised and clarified the legend of Figure 2 to better explain how NPs internalization may trigger cellular stress responses and the activation of the autophagy pathway. In addition, we have expanded the discussion by including a new paragraph (Section 2.7), where we describe in more detail the molecular pathways involved in NP-induced cellular stress and autophagy flux, based on recent literature.

  1. Please check it whether can affect the immune system such as T cell, B cell and DC cells?

We thank the reviewer for the helpful comment. We agree that the effects of NPs on autophagy in immune system cells are interesting and worth investigating. However, at the moment, literature on this specific topic is still lacking. For this reason, we decided to keep a broader view and focus on cell models where more solid data are available, to avoid drawing conclusions that are not well supported yet. Our main goal was to provide an updated and focused overview of how NPs modulate autophagic process in general, and we believe this can still help highlight important gaps and guide future research. We do mention this lack of studies in immune cells briefly in the conclusion section (lines 764-769), to underline the need for more research in that area. Again, we thank the reviewer for the suggestion, which helped us better shape the message of our work.

  1. And please checked it how about in epigenetic effect?

We thank the reviewer for the constructive suggestion. In response, we have carefully examined the potential epigenetic effects induced by nanoplastics and addressed this aspect in Section 2.6.5, which focuses on autophagy modulation within the reproductive system (lines 632-649). In this revised section, we have incorporated recent findings that highlight the interplay between nanoplastic exposure, autophagic responses, and epigenetic alterations. By referencing current literature, we aimed to elucidate how epigenetic mechanisms may contribute to the dysregulation of reproductive function in the context of nanoplastic-induced cellular stress.

Reviewer 2 Report

Comments and Suggestions for Authors

In this manuscript, author reviewed the current evidence on the interactions between NPs and autophagic pathways across diverse biological systems. Most of text were well organized and fully described the results of recent researches. However, the manuscript needs major modifications to be considered by International Journal of Molecular Sciences.

Major modification;

  • In the title, the author attempted to present the biological modulation of autophagy by nanoplastics. However, in the actual text, the characteristics and generation process of nanoplastics are unnecessarily described. Therefore, the author should edit and modify the text to suit the purpose presented in the paper. In addition, it is recommended to add significant experimental results and mechanisms of action for them.
  • Also, author should further describe the cellular uptake mechanism of nanoplasctics.
  • It is recommended to compare the effects of microplasctics and nanoplastics on the biological modulation of autophagy.

Minor modifications;

  • Abstracts should be clearly represented Background, Purpose, Main contents, and Conclusions.
  • All abbreviation should be fully described when it firstly appeared. Also, this description should be not repeated in text.
  • References should be corrected according to journal guideline.

Author Response

Reviewer #2 (Technical Comments to the Author): 

In this manuscript, author reviewed the current evidence on the interactions between NPs and autophagic pathways across diverse biological systems. Most of text were well organized and fully described the results of recent researches. However, the manuscript needs major modifications to be considered by International Journal of Molecular Sciences.

Major modification;

  • In the title, the author attempted to present the biological modulation of autophagy by nanoplastics. However, in the actual text, the characteristics and generation process of nanoplastics are unnecessarily described. Therefore, the author should edit and modify the text to suit the purpose presented in the paper. In addition, it is recommended to add significant experimental results and mechanisms of action for them.

We are grateful to the reviewer for this insightful comment. In order to better align the manuscript content with the title, which aims to address the biological modulation of autophagy by nanoplastics, we have revised the structure of the manuscript and reduced descriptive sections on nanoplastic properties that were not essential to the central focus. In particular, we have added a dedicated section (Section 2.7) called as “Molecular signalling pathways linking NPs to autophagy dysregulation”, in which we elucidate the main molecular mechanisms and intracellular signaling pathways involved in the modulation of autophagy upon nanoplastic exposure. Recent findings have demonstrated that nanoplastics—especially polystyrene nanoplastics (PS-NPs)—can modulate autophagy in a context-dependent manner, either by activating it as a cytoprotective response or by impairing autophagic flux, thereby contributing to cellular stress and toxicity. We believe that the inclusion of this mechanistic discussion provides a more consistent and focused framework, reinforcing the scientific objectives presented in the title and enhancing the overall relevance and depth of the manuscript.

  • Also, author should further describe the cellular uptake mechanism of nanoplasctics.

We thank the reviewer for the helpful suggestion. In response, we have further expanded the discussion on the cellular uptake mechanisms of nanoplastics in Section 2.3 (lines 188-204). Particular emphasis has been placed on the critical role of particle size in determining the internalization pathways and cellular interactions. This revised section now provides a more detailed overview of the main mechanisms involved—such as passive diffusion and endocytosis (including clathrin- and caveolin-mediated pathways)—and highlights how reduced particle size enhances the likelihood of cellular uptake, as supported by recent experimental evidence.

  • It is recommended to compare the effects of microplasctics and nanoplastics on the biological modulation of autophagy.

We thank the reviewer for the insightful comment. We completely agree that both MPs and NPs can modulate autophagy, and that a comparative description of their effects on this process may be of interest. However, our review aims to provide a focused and up-to-date overview of the specific effects of NPs in the modulation of autophagy, which remains still emerging compared to MPs. Moreover, it is now well recognized that due to their nanoscale size, NPs exhibit different and specific physicochemical properties. For example, they have a greater surface area, higher reactivity, and enhanced capacity of cellular penetration. These specific properties make their interactions with autophagic pathways unique, thus justifying a specific and dedicated revision of the literature on these nanoparticles. Thus, our review, by providing a targeted analysis of NPs, can effectively contribute to identifying current knowledge gaps and informing future research in this field. However, to address the reviewer’s point, we briefly justify our choice to focus the current overview of plastic effects on the autophagy process on nanoscale plastics in the introduction section, lines 76-83.

Minor modifications;

  • Abstracts should be clearly represented Background, Purpose, Main contents, and Conclusions.

Thanking you for suggestion. We have corrected.

  • All abbreviation should be fully described when it firstly appeared. Also, this description should be not repeated in text.

Thanking you. We have checked and corrected.

  • References should be corrected according to journal guideline.

Thanking you. We have checked and corrected.

Round 2

Reviewer 2 Report

Comments and Suggestions for Authors

Thank you for thoroughly addressing the reviewers' comments. The revised manuscript has been significantly improved and adequately reflects the suggestions provided. I find the current version acceptable for publication.